# Finding the Needle in the Haystack with Convolutions: on the benefits of architectural bias

**Stéphane d'Ascoli**
stephane.dascoli@ens.fr
Laboratoire de Physique de l'Ecole normale supérieure ENS, Université PSL,
CNRS, Sorbonne Université, Université Paris-Diderot, Sorbonne Paris Cité, Paris, France

**Levent Sagun**
leventsagun@fb.com
Facebook AI Research
Facebook, Paris, France

**Joan Bruna**
bruna@cims.nyu.edu
Courant Institute of Mathematical Sciences and Center for Data Science
New York University, New York City, United States

**Giulio Biroli**
giulio.biroli@lps.ens.fr
Laboratoire de Physique de l'Ecole normale supérieure ENS, Université PSL,
CNRS, Sorbonne Université, Université Paris-Diderot, Sorbonne Paris Cité, Paris, France

## Abstract

Despite the phenomenal success of deep neural networks in a broad range of learning tasks, there is a lack of theory to understand the way they work. In particular, Convolutional Neural Networks (CNNs) are known to perform much better than Fully-Connected Networks (FCNs) on spatially structured data: the architectural structure of CNNs benefits from prior knowledge on the features of the data, for instance their translation invariance. The aim of this work is to understand this fact through the lens of dynamics in the loss landscape.

We introduce a method that maps a CNN to its equivalent FCN (denoted as eFCN). Such an embedding enables the comparison of CNN and FCN training dynamics directly in the FCN space. We use this method to test a new training protocol, which consists in training a CNN, embedding it to FCN space at a certain "relax time", then resuming the training in FCN space. We observe that for all relax times, the deviation from the CNN subspace is small, and the final performance reached by the eFCN is higher than that reachable by a standard FCN of same architecture. More surprisingly, for some intermediate relax times, the eFCN outperforms the CNN it stemmed, by combining the prior information of the CNN and the expressivity of the FCN in a complementary way. The practical interest of our protocol is limited by the very large size of the highly sparse eFCN. However, it offers interesting insights into the persistence of architectural bias under stochastic gradient dynamics. It shows the existence of some rare basins in the FCN loss landscape associated with very good generalization. These can only be accessed thanks to the CNN prior, which helps navigate the landscape during the early stages of optimization.

# 1   Introduction

In the classic dichotomy between model-based and data-based approaches to solving complex tasks, Convolutional Neural Networks (CNN) correspond to a particularly efficient tradeoff. CNNs capture key geometric prior information for spatial/temporal tasks through the notion of local translation invariance. Yet, they combine this prior with high flexibility, that allows them to be scaled to millions of parameters and leverage large datasets with gradient-descent learning strategies, typically operating in the 'interpolating' regime, i.e. where the training data is fit perfectly.

Such regime challenges the classic notion of model selection in statistics, whereby increasing the number of parameters trades off bias by variance [38]. On the one hand, several recent works studying the role of optimization in this tradeoff argue that model size is not always a good predictor for overfitting [30, 38, 29, 18, 7], and consider instead other complexity measures of the function class, which favor CNNs due to their smaller complexity [14]. On the other hand, authors have also considered geometric aspects of the energy landscape, such as width of basins [24], as a proxy for generalisation. However, these properties of the landscape do not appear to account for the benefits associated with specific architectures. Additionally, considering the implicit bias due to the optimization scheme [35, 20] is not enough to justify the performance gains without considering the architectural bias. Despite the important insights on the role of over-parametrization in optimization [13, 3, 36], the architectural bias prevails as a major factor to explain good generalization in visual classification tasks – over-parametrized CNN models generalize well, but large neural networks without any convolutional constraints do not.

In this work, we attempt to further disentangle the bias stemming from the architecture and the optimization scheme by showing that the CNN prior plays a favorable role mostly at the *beginning* of optimization. Geometrically, the CNN prior defines a low-dimensional subspace within the space of parameters of generic Fully-Connected Networks (FCN) (this subspace is linear since the CNN constraints of weight sharing and locality are linear, see Figure 1 for a sketch of the core idea). Even though the optimization scheme is able to minimize the training loss with or without the constraints (for sufficiently over-parametrized models [19, 38]), the CNN subspace provides a "better route" that navigates the loss landscape to solutions with better generalization performance.

Yet, surprisingly, we observe that leaving this subspace at an appropriate time can result in a FCN with an equivalent or even better generalization than a CNN. Our numerical experiments suggest that the CNN subspace *as well as* its vicinity are good candidates for high-performance solutions. Furthermore, we observe a threshold distance from the CNN space beyond which the performance drops back down to the vanilla FCN accuracy level. Our results offer a new perspective on the success of the convolutional architecture: within FCN loss landscapes there exist rare basins associated to very good generalization, characterised not only by their width but rather by their distance to the CNN subspace. These can be accessed thanks to the CNN prior, and are otherwise missed in the usual training of FCNs.

The rest of the paper is structured as follows. Section 2 discusses prior work in relating architecture and optimization biases. Section 3 presents our CNN to FCN embedding algorithm and training procedure, and Section 4 describes and analyses the experiments performed on the CIFAR-10 dataset [25]. We conclude in Section 5 by describing theoretical setups compatible with our observations and consequences for practical applications.

# 2   Related Work

The relationship between CNNs and FCNs is an instance of trading-off prior information with expressivity within Neural Networks. There is abundant literature that explored the relationship between different neural architectures, for different purposes. One can roughly classify these works on whether they attempt to map a large model into a smaller one, or vice-versa.

In the first category, one of the earliest efforts to introduce structure within FCNs with the goal of improving generalization was Nowlan and Hinton's soft weight sharing networks [32], in which the weights are regularized via a Mixture of Gaussians. Another highly popular line of work attempts to *distill* the "knowledge" of a large model (or an ensemble of models) into a smaller one [8, 22, 4], with the goal of improving both computational efficiency and generalization performance. Network

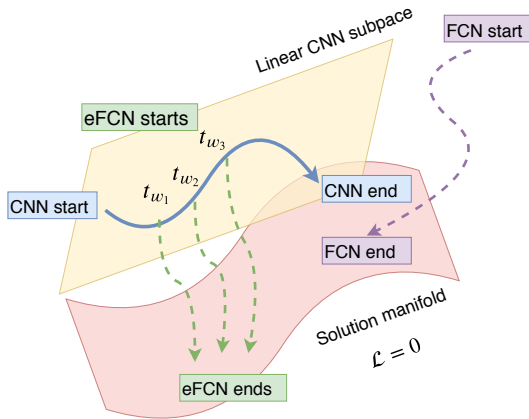

Figure 1: **White background:** ambient, $M$-dimensional, fully-connected space. **Yellow subspace:** linear, $m$-dimensional convolutional subspace. We have $m \ll M$. **Red manifold:** (near-) zero loss valued, (approximate-) solution set for a given training data. Note that it is a nontrivial manifold due to continuous symmetries (also, see the related work section on mode connectivity) and it intersects with the CNN subspace. **Blue path:** a CNN initialized and trained with the convolutional constraints. **Purple path:** a FCN model initialized and trained without the constraints. **Green paths:** Snapshots taken along the CNN training that are lifted to the ambient FCN space, and trained in the FCN space without the constraints.

pruning [21] and the recent "Lottery Ticket Hypothesis" [15] are other remarkable instances of the benefits of model reduction.

In the second category, which is more directly related to our work, authors have attempted to build larger models by embedding small architectures into larger ones, such as the Net2Net model [10] or more evolved follow-ups [34]. In these works, however, the motivation is to accelerate learning by some form of knowledge transfer between the small model and the large one, whereas our motivation is to understand the specific role of architectural bias in generalization.

In the infinite-width context, [31] study the role of translation equivariance of CNNs compared to FCNs. They find that in this limit, weight sharing does not play any role in the Bayesian treatment of CNNs, despite providing significant improvment in the finite-channel setup.

The links between generalization error and the geometry and topology of the optimization landscape have been also extensively studied in recent times. [14] compare generalisation bounds between CNNs and FCNs, establishing a sample complexity advantage in the case of linear activations. [28, 27] obtain specific generalisation bounds for CNN architectures. [9] proposed a different optimization objective, whereby a bilateral filtering of the landscape favors dynamics into wider valleys. [24] explored the link between sharpness of local minima and generalization through Hessian analysis [33], and [37] argued in terms of the volume of basins of attraction. The characterization of the loss landscape along paths connecting different models have been studied recently, e.g. in [16], [17], and [12]. The existence of rare basins leading to better generalization was found and highlighted in simple models in [5, 6]. The role of the CNN prior within the ambient FCNs loss landscape and its implication for generalization properties were not considered in any of these works. In the following we address this point by building on these previous investigations of the landscape properties.

## 3 CNN to FCN Embedding

In both FCNs and CNNs, each feature of a layer is calculated by applying a non-linearity to a weighted sum over the features of the previous layer (or over all the pixels of the image, for the first layer). CNNs are a particular type of FCNs, which make use of two key ingredients to reduce their number of redundant parameters: locality and weight sharing.

*Locality:* In FCNs, the sum is taken over all the features of the previous layer. In locally connected networks (LCNs), locality is imposed by restricting the sum to a small receptive field (a box of adjacent features of the previous layer). The set of weights of this restricted sum is called a filter. For a given receptive field, one may create multiple features (or channels) by using several different filters. This procedure makes use of the spatial structure of the data and reduces the number of fitting parameters.

*Weight sharing:* CNNs are a particular type of LCNs where all the filters of a given channel use the same set of weights. This procedure makes use of the somewhat universal properties of feature

extracting filters such as edge detectors and reduces even more drastically the number of fitting parameters.

When mapping a CNN to its equivalent FCN (eFCN), we obtain very sparse (due to locality) and redundant (due to weight sharing) weight matrices (see Sec. A of the Supplemental Material for some intuition on the mapping). This typically results in a large memory overhead as the eFCN of a simple CNN can take several orders of magnitude more space in the memory. Therefore, we present the core ideas on a simple 3-layer CNN on CIFAR-10 [26], and show similar results for AlexNet on CIFAR-100 in Sec. B of the Supplemental Material.

In the mapping[1], all layers apart form the convolutional layers (ReLU, Dropout, MaxPool and fully-connected) are left unchanged except for proper reshaping. Each convolutional layer is mapped to a fully-connected layer.

*As a result, for a given CNN, we obtain its eFCN counterpart with an end-to-end fully-connected architecture which is functionally identical to the original CNN.*

## 4 Experiments

We are given input-label pairs for a supervised classification task, $(x, y)$, with $x \in \mathbb{R}^d$ and $y$ the index of the correct class for a given image $x$. The network, parametrized by $\theta$, outputs $\hat{y} = f_x(\theta)$. To distinguish between different architectures we denote the CNN weights by $\theta^{CNN} \in \mathbb{R}^m$ and the eFCNs weights by $\theta^{eFCN} \in \mathbb{R}^M$. Let's denote the embedding function described in Sec. 3 by $\Phi : \mathbb{R}^m \mapsto \mathbb{R}^M$ where $m \ll M$ and with a slight abuse of notation use $f(\cdot)$ for both CNN and eFCN. Dropping the explicit input dependency for simplicity we have:

$$f(\theta^{CNN}) = f(\Phi(\theta^{CNN})) = f(\theta^{eFCN}).$$

For the experiments, we prepare the CIFAR-10 dataset for training without data augmentation. The optimizer is set to stochastic gradient descent with a constant learning rate of 0.1 and a minibatch size of 250. We turn off the momentum and weight decay to simply focus on the stochastic gradient dynamics and we do not adjust the learning rate throughout the training process. In the following, we focus on a convolutional architecture with 3 layers, 64 channels at each layer that are followed by ReLU and MaxPooling operators, and a single fully connected layer that outputs prediction probabilities. In our experience, this VanillaCNN strikes a good balance of simplicity and performance in that its equivalent FCN version does not suffer from memory issues yet it significantly outperforms any FCN model trained from scratch. We study the following protocol:

1. Initialize the VanillaCNN at $\theta_{init}^{CNN}$ and train for 150 epochs. At the end of training $\theta_{final}^{CNN}$ reaches $\sim 72\%$ test accuracy.
2. Along the way, save $k$ snapshots of the weights at logarithmically spaced epochs: $\{t_0 = 0, t_1, \ldots, t_{k-2}, t_{k-1} = 150\}$. It provides $k$ CNN points denoted by $\{\theta_{t_0}^{CNN} = \theta_{init}^{CNN}, \theta_{t_1}^{CNN}, \ldots, \theta_{t_{k-1}}^{CNN}\}$.
3. Lift each one to its eFCN: $\{\Phi(\theta_{t_0}^{CNN}), \ldots, \Phi(\theta_{t_{k-1}}^{CNN})\} = \{\theta_{t_0}^{eFCN}, \ldots, \theta_{t_{k-1}}^{eFCN}\}$ (so that only $m$ among a total of $M$ parameters are non-zero).
4. Train these $k$ eFCNs in the FCN space for 100 epochs in the same conditions, except a smaller learning rate of 0.01. We obtain $k$ solutions $\{\theta_{t_0,final}^{eFCN}, \ldots, \theta_{t_{k-1},final}^{eFCN}\}$.
5. For comparison, train a standard FCN (with the same architecture as the eFCNs but with the default PyTorch initialization) for 100 epochs in the same conditions as the eFCNs, and denote the resulting weights by $\theta_{final}^{FCN}$. The latter reaches $\sim 55\%$ test accuracy.

This process gives us one CNN solution, one FCN solution, and $k$ eFCN solutions that are labeled as

$$\theta_{final}^{CNN}, \theta_{final}^{FCN}, \text{ and } \{\theta_{t_0,final}^{eFCN}, \ldots, \theta_{t_{k-1},final}^{eFCN}\} \tag{1}$$

which we analyze in the following subsections. Note that due to the difference in size between the CNN and the eFCNs, it unclear what learning rate would give a fair comparison. One solution, shown in Sec. B of the Supplemental Material, is to use an adaptive learning rate optimizer such as Adam.

### 4.1 Performance and training dynamics of eFCNs

Our first aim is to characterize the training dynamics of eFCNs and study how their training evolution depends on their *relax time* $t_w \in \{t_0 = 0, t_1, \ldots, t_{k-2}, t_{k-1} = 150\}$ (in epochs). When the architectural constraint is relaxed, the loss decreases monotonically to zero (see the left panel of Fig. 2). The initial losses are smaller for larger $t_w$s, as expected since those $t_w$s correspond to CNNs trained for longer. In the right panel of Fig. 2, we show a more surprising result: test accuracy increases monotonously in time for all $t_w$s, thus showing that *relaxing the constraints does not lead to overfitting or catastrophic forgetting.* Hence, from the point of view of the FCN space, it is not as if CNN dynamics took place on an unstable region from which the constraints of locality and weight sharing prevented from falling off. It is quite the contrary instead: the CNN dynamics takes place in a basin, and when the constraints are relaxed, the system keeps going down on the training surface and up in test accuracy, as opposed to falling back to the standard FCN regime.

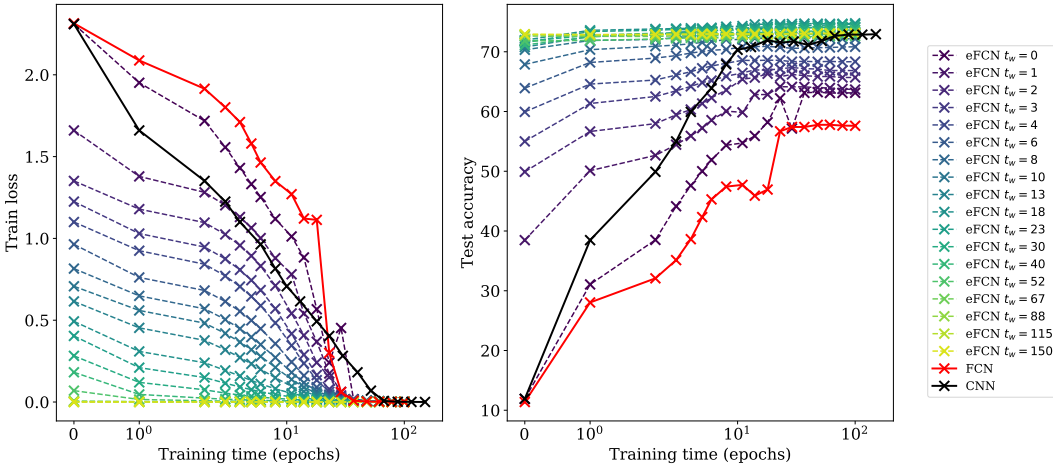

Figure 2: Training loss (**left**) and test accuracy (**right**) on CIFAR-100 vs. training time in logarithmic scale including the initial point. Different models are color coded as follows: the VanillaCNN is shown in black, standard FCN is in red, and the eFCNs with their relax time $t_w$s are indicated by the gradient ranging from purple to light green.

In Fig. 3 (left) we compare the final test accuracies reached by eFCN with the ones of the CNN and the standard FCN. We find two main results. First, the accuracy of the eFCN for $t_w = 0$ is approximately at $62.5\%$, well above the standard FCN result of $57.5\%$. This shows that imposing an *untrained* CNN prior is already enough to find a solution with much better performance than a standard FCN. Hence the CNN prior brings us to a good region of the landscape *to start with*. The second result, perhaps even more remarkable, is that at intermediate relax times ($t_w \sim 20$ epochs), the eFCN reaches—and exceeds—the final test accuracy reached by the CNN it stemmed from. This supports the idea that the constraints are mostly helpful for navigating the landscape *during the early stages of optimization*. At late relax times, the eFCN is initialized close to the bottom of the landscape and has little room to move, hence the test accuracy stays the same as that of the fully trained CNN.

### 4.2 A closer look at the landscape

A widespread idea in the deep learning literature is that the sharpness of the minima of the training loss is related to generalization performance [24, 23]. The intuition being that flat minima reduce the effect of the difference between training loss and test loss. This motivates us to compare the first and second order properties of the landscape explored by the eFCNs and the CNNs they stem from. To do so, we investigate the norm of the gradient of the training loss, $|\nabla \mathcal{L}|$, and the top eigenvalue of the Hessian of the training loss, $\lambda_{max}$, in the central and right panels of Fig. 3 (we calculate the latter using a power method).

We point out several interesting observations. First, the sharpness ($|\nabla \mathcal{L}|$) and steepness ($\lambda_{max}$) indicators increase then decrease during the training of the CNN (as analyzed in [1]), and display a maximum around $t_w \simeq 20$, which coincides with the relax time of best improvement for the eFCNs.

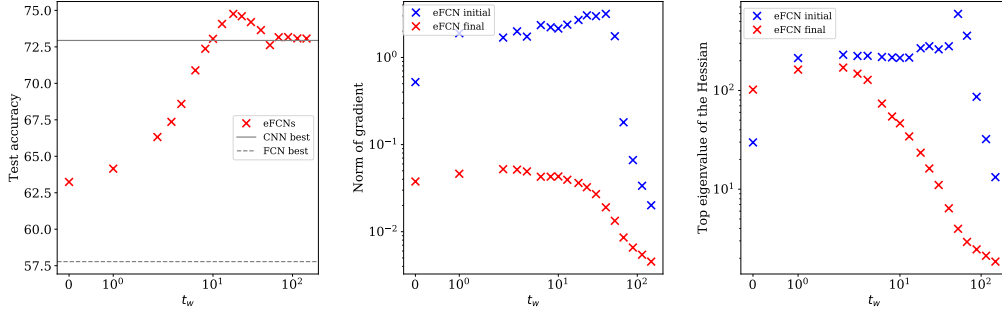

Figure 3: **Left**: Performance of eFCNs reached at the end of training (red crosses) compared to its counterpart for the best CNN accuracy (straight line) and the best FCN accuracy (dashed line). **Center**: Norm of the gradient for eFCNs at the beginning and at the end of training. **Right**: Largest eigenvalue of the Hessian for eFCNs at the beginning and at the end of training. In all figures the $x$-axis is the relax time $t_w$.

Second, we see that after training the eFCNs, these indicators plummet by an order of magnitude, which is particularly surprising at very late relax time where it appeared in the left panel of Fig. 3 (see also 4) as if the eFCNs was hardly moving away from initialization. This supports the idea that when the constraints are relaxed, the extra degrees of freedom *lead us to wider basins*, possibly explaining the gain in performance.

### 4.3 How far does the eFCN escape from the CNN subspace?

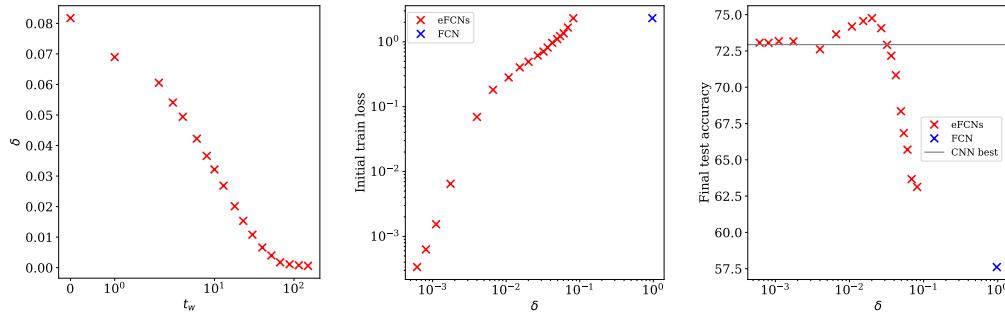

Figure 4: **Left panel:** relax time $t_w$ of the eFCN vs. $\delta$, the measure of deviation from the CNN subspace through the locality constraint, at the final point of eFCN training. **Middle panel:** $\delta$ vs. the initial loss value. **Right panel:** $\delta$ vs. final test accuracy of eFCN models. For reference, the blue point in the **middle** and **right** panels indicate the deviation measure for a standard FCN, where $\delta \sim 97\%$.

A major question naturally arises: how far do the eFCNs move away from their initial condition? Do they stay close to the sparse configuration they were initialized in ? To answer this question, we quantify how locality is violated once the constraints are relaxed (violation of weight sharing will be studied in Sec. 4.4). To this end, we consider a natural decomposition of the weights in the FCN space into two parts, $\theta = (\theta_{\text{local}}, \theta_{\text{off-local}})$, where $\theta_{\text{off-local}} = 0$ for an eFCN when it is initialized from a CNN. A visualization of these blocks may be found in Sec. A of the Supplemental Material. We then study the ratio $\delta$ of the norm of the off-local weights to the total norm, $\delta(\theta) = \frac{||\theta_{\text{off-local}}||_2}{||\theta||_2}$, which is a measure of the deviation of the model from the CNN subspace.

Fig. 4 (left) shows that the deviation $\delta$ at the end of eFCN training decreases monotonically with its relax time $t_w$. Indeed, the earlier we relax the constraints (and therefore the higher the initial loss of the eFCN) the further the eFCN escapes from the CNN subspace, as emphasized in Fig. 4 (middle). However, even at early relax times, the eFCNs stay rather close to the CNN subspace, since the ratio

never exceeds 8%, whereas it is around 97% for a regular FCN (since the number of off-local weights is much larger than the number of local weights). This underlines the *persistence of the architectural bias under the stochastic gradient dynamics*.

Fig. 4 (right) shows that when we move away from the CNN subspace, performance stays high then plummets down to FCN level. *This hints to a critical distance from the CNN subspace within which eFCNs behave like CNNs, and beyond which they fall back to the standard FCN regime.* We further explore this high performance vicinity of the CNN subspace using interpolations in weight space in Sec. C of the Supplemental Material.

### 4.4 What role do the extra degrees of freedom play in learning?

How can the eFCN use the extra degrees of freedom to improve performance ? From Fig. 5, we see that the off-local part of the eFCN is useless on its own (with the local part masked off). However, when combined with the local part, it may greatly improve performance when the constraints are relaxed early enough. This hints to the fact that the local and off-local parts are performing complementary tasks.

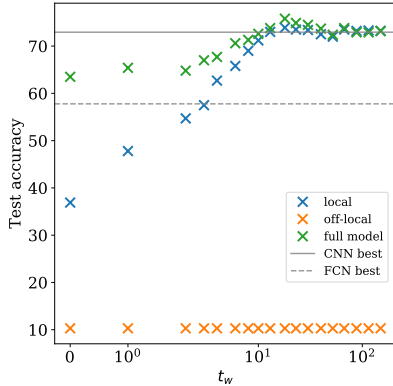

To understand what tasks the two parts they are performing, we show in Fig. 6 a "filter" from the first layer of the eFCN (whose receptive field is of the size of the images since locality is relaxed). Note that each CNN filter gives rise to many eFCN filters : one for each position of the CNN filter on the image, since weight sharing is relaxed. Here we show the one obtained when the CNN filter (local block) is on the top left of the image. We see that off-local blocks stay orders of magnitude smaller than the local blocks, as expected from Sec. 4.3 where we saw that locality was almost conserved. We also see that local blocks hardly change during training, showing that weight sharing of the local blocks is also almost conserved.

Figure 5: Contributions to the test accuracy of the local blocks (off-local blocks masked out), in orange, and off-local blocks (local blocks masked out), in blue. Combining them together yields a large gain in performance for the eFCN, in green.

More surprisingly, we see that for $t_w > 0$ distinctive shapes of the images are learned by the eFCN off-local blocks, which perform some kind of template-matching. Note that the silhouettes are particularly clear for the intermediate relax time (middle row), at which we know from Sec. 4.1 that the eFCN had the best improvement over the CNN. *Hence, the eFCN is combining template-matching with convolutional feature extraction in a complementary way.*

Note that by itself, template-matching is very inefficient for complicated and varied images such as those of the CIFAR-10 dataset. Hence it cannot be observed in standard FCNs, as shown in Fig. 7 where we reproduce the counterpart of Fig. 6 for the FCN in the left and middle images (they correspond to initial and final training times respectively). To reveal the silhouettes learned, we need to look at the pixelwise difference between the two images, i.e. focus on the change due to training (this in unnecessary for the eFCN whose off-local weights started at zero). In the right image of Fig. 7), we see that a loose texture emerges, however, it is not as sharp as that of the eFCN weights after training. Template-matching is only useful as a cherry-on-the-cake alongside more efficient learning procedures.

## 5 Discussion and Conclusion

In this work, we examined the inductive bias of CNNs, and challenged the accepted view that FCNs are unable to generalize as well as CNNs on visual tasks. Specifically, we showed that the CNN prior is mainly useful during the early stages of training, to prevent the unconstrained FCN from falling prey of spurious solutions with poor generalization too early.

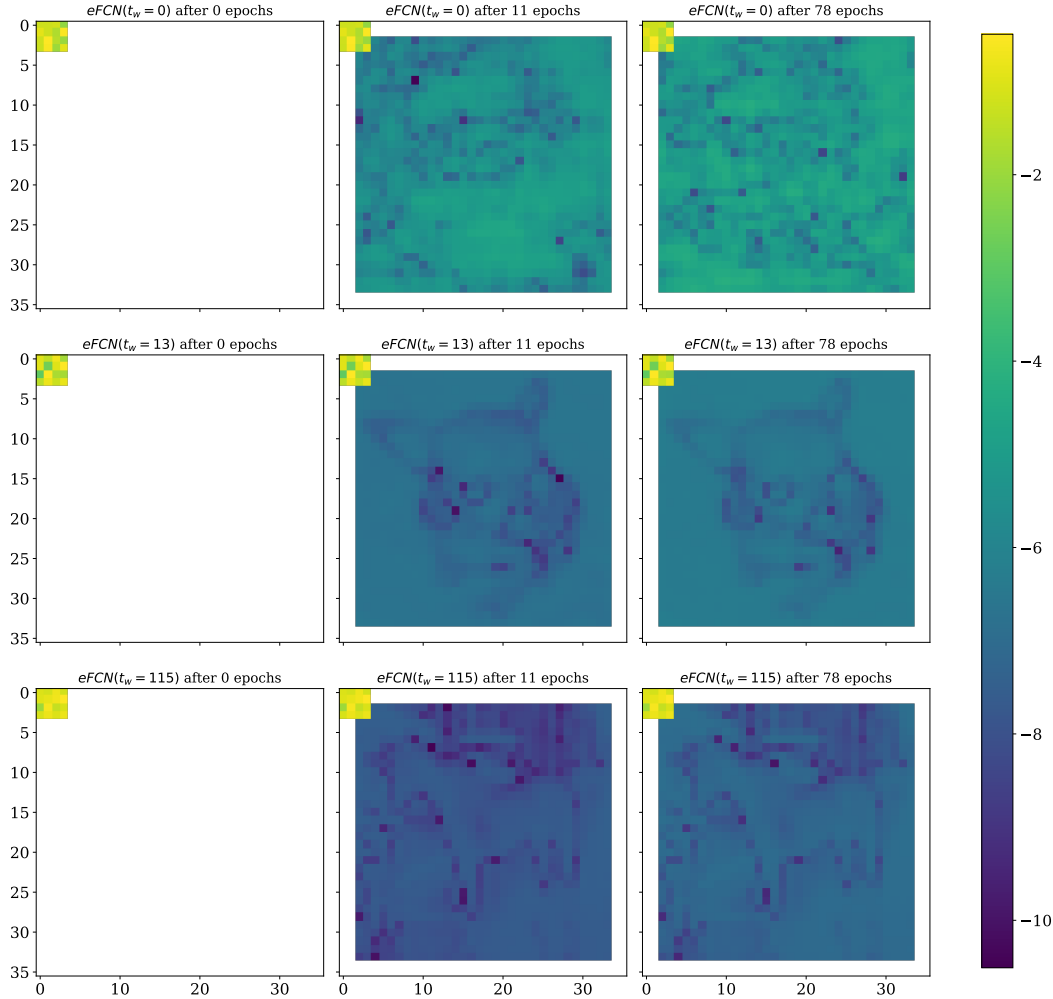

Figure 6: Heatmap of the weights of an eFCN "filter" from the first layer just at relax time (**left** column), after training for 11 epochs (**middle** column), and after training for 78 epochs (**right** column). The eFCNs were initialized at relax times $t_w = 0$ (**top** row), $t_w = 13$ (**middle** row), and $t_w = 115$ (**bottom** row). The colors indicate the natural logarithm of the absolute value of the weights. Note that the convolutional filters, in the top right, vary little and remain orders of magnitude larger than the off-local blocks, whereas the off-local blocks pick up strong signals from images as sharp silhouettes appear.

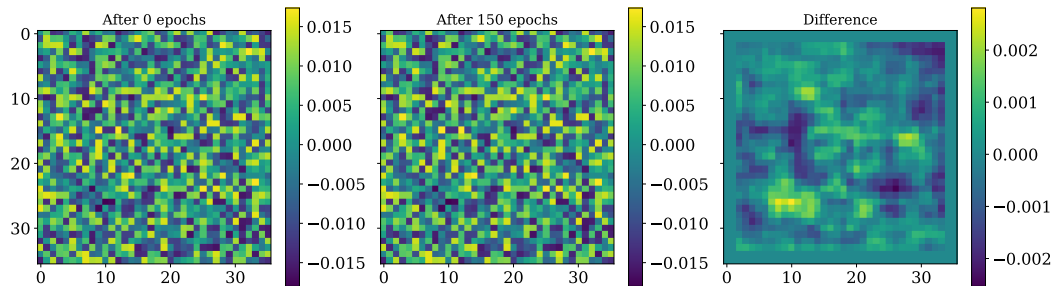

Figure 7: Same heatmap of weights as shown in Fig. 6 but for a standard FCN at a randomly initialized point (**left**) and after training for 150 epochs (**middle**). The pixelwise difference is shown on the **right** panel. A loose texture appears, but it is by no means as sharp as the silhouettes of the eFCNs.

Our experimental results show that there exists a vicinity of the CNN subspace with high generalization properties, and one may even enhance the performance of CNNs by exploring it, if one relaxes the CNN constraints at an appropriate time during training. The extra degrees of freedom are used to perform complementary tasks which alone are unhelpful. This offers interesting theoretical perspectives, in relation to other high-dimensional estimation problems, such as in spiked tensor models [2], where a smart initialization, containing prior information on the problem, is used to provide an initial condition that bypasses the regions where the estimation landscape is "rough" and full of spurious minima.

On the practical front, despite the performance gains obtained, our algorithm remains highly impractical due to the large number of degrees of freedom required on our eFCNs. However, more efficient strategies that would involve a less drastic relaxation of the CNN constraints (e.g., relaxing the weight sharing but keeping the locality constraint such as locally-connected networks [11]) could be of potential interest to practitioners.

**Acknowledgments**

We would like to thank Alp Riza Guler and Ilija Radosavovic for helpful discussions. We acknowledge funding from the Simons Foundation (#454935, Giulio Biroli). JB acknowledges the partial support by the Alfred P. Sloan Foundation, NSF RI-1816753, NSF CAREER CIF 1845360, and Samsung Electronics.

## Footnotes

[1]The source code may be found at: https://github.com/sdascoli/anarchitectural-search.

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
