[Supplementary Material 1 · CNN-FC SupMat.pdf]

# Finding the Needle in the Haystack with Convolutions: on the benefits of architectural bias

## Supplemental Material

**Stéphane d'Ascoli**
stephane.dascoli@ens.fr
Laboratoire de Physique de l'Ecole normale supérieure ENS, Université PSL,
CNRS, Sorbonne Université, Université Paris-Diderot, Sorbonne Paris Cité, Paris, France

**Levent Sagun**
leventsagun@fb.com
Facebook AI Research
Facebook, Paris, France

**Joan Bruna**
bruna@cims.nyu.edu
Courant Institute of Mathematical Sciences and Center for Data Science
New York University, New York City, United States

**Giulio Biroli**
giulio.biroli@lps.ens.fr
Laboratoire de Physique de l'Ecole normale supérieure ENS, Université PSL,
CNRS, Sorbonne Université, Université Paris-Diderot, Sorbonne Paris Cité, Paris, France

## 1 Visualizing the embedding

In Fig. 1, we provide an illustration of the mapping from CNN to eFCN. Denoting as $k, s, p$ the filter size, stride and padding of the convolution, we have the following:

$$d_{in} = 4$$
$$(k, s, p) = (3, 1, 0)$$
$$d_{out} = \frac{d_{in} + 2p - k}{s} + 1 = 2$$

The eFCN layer is of size $(c_{in} \times d_{in} \times d_{in}, c_{out} \times d_{out} \times d_{out}) = (4, 16)$ since $c_{in} = c_{out} = 1$ here. In Fig. 2, we show the typical structure of the eFCN weight matrices observed in practice.

## 2 Results with AlexNet on CIFAR-100

In this section, we show that the ideas we presented in the main text hold for various classes of data, architecture and optimizer. Namely, we show that our results hold when switching from SGD to Adam on CIFAR10, and for AlexNet [3] on the CIFAR-100 dataset. Each subsection contains figures which are counterparts of the ones of the main text : performance and training dynamics of the eFCNs in Fig. 3, deviation from CNN subspace in Fig. 4, role of off-local blocks in learning in Fig. 5.

Figure 1: eFCN wight matrix (**bottom**) obtained when acting on an input of size of size (4,4) (**top left**) with a filter of size (3,3) (**top right**). The colors of the eFCN weight matrix show where they stem from in the filter (the off-local blocks, in yellow, are set to zero at initialization).

Figure 2: **Top**: Heatmap of a block of weights corresponding to the first input channel and the first output channel of the first layer of the eFCN just after its initialization from the converged VanillaCNN. The colorscale indicates the natural logarithm of the absolute value of the weights. The highly sparse and self-repeating structure of the weight matrix is due to the locality and weight sharing constraints. **Bottom**: Same after training the eFCN for 100 epochs. The off-local blocks appear in blue : their weights are several orders of magnitude smaller in absolute value than those of the local blocks, in yellow. Note that due to the padding many weights stay at zero even after relaxing the constraints. When unflattened, the first row of this heatmap gives rise to the images shown in Fig. 5.

Figure 3: This figure sums up in a compact way the generalization dynamics of the eFCNs. The red curve represents the test accuracy of the model versus its training time in epochs. Above each point $t_w$ of the training, we depict as crosses the test accuracy history of the eFCN stemmed at relax time $t_w$, with colors indicating the training time of the eFCN after embedding. For comparison, the best test accuracy reached by a standard FCN of same size is depicted as a brown horizontal dashed line. **Left**: VanillaCNN on CIFAR-10, with Adam optimizer. **Right**: Alexnet on CIFAR-100, with SGD optimizer. We note that results are qualitatively similar : the eFCNs always improve after initialization, outperform the standard FCN, and we again observe that for some relax times the eFCNs even exceeds the best test accuracy reached by the CNN.

Figure 4: **Left panel:** relax time $t_w$ of the eFCN vs. $\delta$, the measure of deviation from the CNN subspace through the locality constraint, at the final point of eFCN training. **Middle panel:** $\delta$ vs. the initial loss value. **Right panel:** $\delta$ vs. final test accuracy of eFCN models. For reference, the blue point in the **middle** and **right** panels indicate the deviation measure for a standard FCN, where $\delta \sim 97\%$.

# 3 Interpolating between CNNs and eFCNs

Another way to understand the dynamics of the eFCNs is to examine the paths that connect them to the CNN they stemmed from in the FCN weight space. Interpolating in the weight space has received some attention in recent literature, in papers such as [1, 2], where it has been shown that contrary to previous beliefs the bottom of the landscapes of deep neural networks resembles a flat, connected level set since one can always find a path of low energy connecting minima.

Here we use two interpolation methods in weight space. The first method, labeled "linear", consists in sampling $n$ equally spaced points along the linear path connecting the weights. Of course, the interpolated points generally have higher training loss than the endpoints.

The second method, labeled "string", consists in starting from the linear interpolation path, and letting the interpolated points fall down the landscape following gradient descent, while ensuring that they stay close enough together by adding an elastic term in the loss :

$$\mathcal{L}_{elastic} = \frac{1}{2}k\sum_{i=1}^{n-1}(\mathbf{x_{i+1}} - \mathbf{x_i})^2 \tag{1}$$

By adjusting the stiffness constant $k$ we can control how straight the string is: at high $k$ we recover the linear interpolation, whereas at low $k$ the points decouple and reach the bottom of the landscape, but are far apart and don't give us an actual path. Note that this method is a simpler form of the one used in [1], where we don't use the "nudging" trick.

For comparison, we also show the performance obtained when interpolating directly in output space (as done in ensembling methods).

Results are shown in figure 6, with the $x$-axis representing the interpolation parameter $\alpha \in [0, 1]$. We see that for both the linear and string interpolations, the training loss profile displays a barrier, except at late $t_w$ where the the eFCN has not escaped far from the CNN subspace. Although the string method fails to find a path without a barrier, this is not sufficient to conclude that no paths exist.

However, the behavior of test accuracy is much more surprising. In all cases, despite the increase in training loss, the interpolated paths reach higher test accuracies than the endpoints, even at early $t_w$ when the eFCN and the CNN are quite far from each other. This confirms that there is a basin of high generalization around the CNN subspace, and that optimum performance can actually be found somewhere in between the solution found by the CNN and the solution found by the eFCN. This offers yet another procedure to improve the performance in practice. However, in all cases we note that the gain in accuracy is lower than the gain obtained by interpolating in output space.

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

Figure 5: **Left**: Visualization of an eFCN "filter" from the the first layer just after embedding (left column), after training after 11 epochs (middle column), and training after 78 epochs (right column); where the eFCN is initialized at relax times $t_w = 0$ (top row), $t_w = 13$ (middle row), and $t_w = 115$ (bottom row). The colors indicate the natural logarithm of the absolute value of the weights. **Right**: Contributions to the test accuracy of the local blocks (off-local blocks masked out) and off-local blocks (local blocks masked out).

(a) $t_w = 0$

(b) $t_w = 5$

(c) $t_w = 18$

(d) $t_w = 61$

Figure 6: Interpolation between the solution reached by the CNN after 100 epochs (interpolation parameter $\alpha = 0$) and the solution found by the eFCN after 100 epochs (interpolation parameter $\alpha = 1$), for four different relax times $t_w$ indicated below the subfigures. In each subfigure, the **left** panel shows train loss, and the **right** panel shows test accuracy. The orange line corresponds to linear interpolation, the blue line corresponds to string method interpolation, and the green line corresponds to interpolation in output space.



[Supplementary Material 2]



Legend:
- CNN train
- FCN best train ac[...]
- × eFCN $t_w = 0$
- × eFCN $t_w = 1$
- × eFCN $t_w = 2$
- × eFCN $t_w = 3$
- × eFCN $t_w = 4$
- × eFCN $t_w = 6$
- × eFCN $t_w = 8$
- × eFCN $t_w = 10$
- × eFCN $t_w = 13$
- × eFCN $t_w = 18$
- × eFCN $t_w = 23$
- × eFCN $t_w = 30$
- × eFCN $t_w = 40$
- × eFCN $t_w = 52$
- × eFCN $t_w = 67$
- × eFCN $t_w = 88$
- × eFCN $t_w = 115$
- × eFCN $t_w = 150$

Axis labels: Accuracy (y-axis), Time in epochs (x-axis)

[Supplementary Material 3]



Legend:
- eFCN $t_w = 0$
- eFCN $t_w = 1$
- eFCN $t_w = 2$
- eFCN $t_w = 3$
- eFCN $t_w = 4$
- eFCN $t_w = 6$
- eFCN $t_w = 8$
- eFCN $t_w = 10$
- eFCN $t_w = 13$
- eFCN $t_w = 18$
- eFCN $t_w = 23$
- eFCN $t_w = 30$
- eFCN $t_w = 40$
- eFCN $t_w = 52$
- eFCN $t_w = 67$
- eFCN $t_w = 88$
- eFCN $t_w = 115$
- eFCN $t_w = 150$
- FCN
- CNN

Left plot axes: Train loss vs Training time (epochs)

Right plot axes: Test accuracy vs Training time (epochs)

[Supplementary Material 4]



Legend:
- CNN train
- FCN best train acc
- eFCN $t_w = 0$
- eFCN $t_w = 1$
- eFCN $t_w = 2$
- eFCN $t_w = 3$
- eFCN $t_w = 4$
- eFCN $t_w = 6$
- eFCN $t_w = 8$
- eFCN $t_w = 10$
- eFCN $t_w = 13$
- eFCN $t_w = 18$
- eFCN $t_w = 23$
- eFCN $t_w = 30$
- eFCN $t_w = 40$
- eFCN $t_w = 52$
- eFCN $t_w = 67$
- eFCN $t_w = 88$
- eFCN $t_w = 115$
- eFCN $t_w = 150$

Axis labels: Accuracy (y-axis), Time in epochs (x-axis)

[Supplementary Material 5]



Legend:
- CNN test
- FCN best test acc
- eFCN $t_w = 0$
- eFCN $t_w = 1$
- eFCN $t_w = 2$
- eFCN $t_w = 3$
- eFCN $t_w = 4$
- eFCN $t_w = 6$
- eFCN $t_w = 8$
- eFCN $t_w = 10$
- eFCN $t_w = 13$
- eFCN $t_w = 18$
- eFCN $t_w = 23$
- eFCN $t_w = 30$
- eFCN $t_w = 40$
- eFCN $t_w = 52$
- eFCN $t_w = 67$
- eFCN $t_w = 88$
- eFCN $t_w = 115$
- eFCN $t_w = 150$

Axis labels: Accuracy (y-axis), Time in epochs (x-axis)