[Reviews · NeurIPS 2019]

Reviewer 1



Post author response: I have read the author's response and other reviews. I believe author's contribution in understanding architectural bias merits publication. My overall score remains the same. Additional comment: is AlexNet on CIFAR-100 without data augmentation? Besides being harder than CIFAR-10, that may explain why 40% accuracy seems lower than other reported numbers on CIFAR-100. Authors should either clarify this fact; or even more interesting show effects don't change in the presence of data augmentation. -------------------------------------------------------------------- Author’s study architectural bias of convolutional networks compared to the fully connected networks. For many interesting tasks, we’ve observed that CNN perform much better than FC networks. Although we understand this superior performance from CNNs incorporating prior knowledge of structured data, understanding of the training dynamics at the level of loss landscape is quite low. By mapping CNN to equivalent FCN(eFCN) and studying training dynamics of eFCN, the authors provide a new tool to investigate CNN architectural bias in loss landscape. Although the practicality of proposed method is probably low (finding efficient way to relax locality constraint would be an interesting future work as suggested by the authors), the utility of the method is quite novel and and shed light on SGD training dynamics of different architectures. The paper is very clearly written with proposed ideas and methods are easy to follow. The experimental results are presented in a logical and understandable manner. Proposed method is simple to understand but yet quite novel, as far as I could tell, providing unknown insights of GD based training on CNNs and FCNs. The paper provides significant novel insights through their new proposed method. Few interesting results observed 1) CNN initialization in the ambient FC space provide better model than FC init on original space 2) Some intermediate switch time after CNN init to ambient eFC space can find better model than full CNN model. 3) Sharpness indicators (gradient norm/max eigenvalue of Hessian) are large during that intermediate switch time and then becomes quite small trained in eFCN space. In the supplementary material, the authors show that the findings also show on different architecture and dataset. Also they provide codes to reproduce results which will allow researchers to build on the findings to gain more insights in CNN architectural bias regards to FCN.

Reviewer 2



The authors show how a CNN prior (local and sharing constraints on the weights) on FCN weights can find better local minima compared to FCN with no constraints. This is suggesting that the architectural bias is only required in the initial part of the optimization to avoid trivial local minima which don't generalize better. One interesting observation is that even with an initial CNN prior FCN tends to performs quite well compared to regular FCN. The experimental results also suggest that a combination of template matching and local filters tend to give better performance compared to only template matching or only local filters. Many interesting insights presented based on experimental validation only.Perhaps either a more thorough experimental analysis or some theoretical evidence can be provided. If the authors claim achieving better performance by relaxing constraints at right point during training, it needs more experimental validation.

Reviewer 3



This is an empirical study to understand why over-parametrized CNN performs better than a Fully Connected Network. As fas as I know, on one did the same experiments before. The results suggested that the architectural bias is not necessary through all the training pass. My key concern about this paper was the experiments. In the paper, it did on a realistic task (cifar), but the model looks small. It's unclear if the model has more parameters or more advanced architecture, does the conclusion still holds? It become even more suspicious, if we look at the results on supplementary material. One AlexNet, the accuracy is only 40%. How come AlexNet performs so bad? If I remember correctly, even FCN can reach 70% with carefully tuning. In another word, there are tons of CNN which performs good on Cifar-10, why pick AlexNet? It's also unclear how the optimization approach affect the results. The paper use a smaller learning rate (0.01) to fine-tune. Is that sensitive or not? Also, it's unclear if the CNN also fine-tuned by this smaller learning rate. If CNN always use the constant learning rate, it doesn't looks fair to me. The paper didn't cite "Do Deep Nets Really Need to be Deep?" which use a FCN to mimic a CNN. Overall, I feel it's an interesting study but the experiments do not strong enough to support the conclusion.

[Author Response · NeurIPS 2019]

We thank the reviewers for their attention and constructive feedback that will improve the quality of our work.

**Reviewer 1:** We acknowledge the need for clearer referencing to the SM and will improve this. Also, the study of
Novak et al. 2019 on the role of locality is indeed relevant; we will discuss it in the revised version.

We agree that studying the effect of further training methods and architectures would be very interesting. Such
exploration is certainly an exciting direction for new research. One practical obstacle for different architectures lies in
the implementation of the mapping for each layer. Our work presents a proof of concept that will hopefully trigger
several investigations along the lines proposed by the reviewer.

One step we took in this direction, motivated by reviewers' suggestions, is to replicate the same experiment with
Adam instead of fixed learning rate SGD. This additional result, which we will add in the revised version, confirms the
phenomena shown in Figure 1 of the SM.

Further subjects we would like to study in a new paper: the effects of locality and weight sharing; embedding of CNNs
into other CNNs with bigger filters; explore the effects of data augmentation on the off-diagonal blocks; scale the
experiments to near SOTA models on CIFAR10; and more...

**Reviewer 2:** We did our best to motivate the fact that relaxing the constraints at the right point is a promising training
technique, by showing the performance improvement in two simple setups on CIFAR-10 and CIFAR-100. To strengthen
our claims, we are working on implementing a softer constraint relaxation by mapping to locally-connected space rather
the fully-connected space. This approach has stronger practical benefits as the increase in the model size is much less
than the eFCN embedding.

In the revised version we will make clearer the potential implications of our method to practitioners, and its foreseeable
extensions. We agree that this emphasis will be complementary to the study of the effects of architectural bias in and of
itself.

**Reviewer 3:** We acknowledge that the "VanillaCNN" model used on CIFAR-10 is rather small, nevertheless its
generalization performance is almost the same as simplified AlexNet on CIFAR-10. Our strategy has been to present the
results for the VanillaCNN in the main text and then validate them for more realistic setups in the SM. The reviewer may
have overlooked that the experiment with AlexNet is performed on CIFAR-100 (not CIFAR-10 as for the VanillaCNN),
which is why the test accuracy is $\sim 40\%$. Furthermore, in our experience, the best tuned FCN on CIFAR-10 hardly
beats $\sim 60\%$. The main reason we present the VanillaCNN in the main text is that it is practically unfeasible to perform
the Hessian analysis on AlexNet (with our computational constraints). In the revised version we will clarify better the
link between the results presented in the main text and the SM.

We understand the very valid concern of the reviewer about the learning rate scheduling, and had actually considered this question, although we do not discuss it in the main text. If our understanding is correct, the reviewer suspects that the fact that the learning rate is divided by 10 upon switching gives the eFCN an unfair advantage over the CNN which keeps a constant learning rate. However, learning rates are intrinsically related to model sizes, and considering how different the CNN and the eFCN are in size, it would be a tricky (yet interesting) question to define what would be a "fair" learning rate to use for the eFCNs. Therefore we solely chose learning rates of 0.1 for the CNNs and 0.01 for the eFCNs so that the corresponding models would all converge in a reasonable and comparable timescale of the order of 100 epochs.

Figure 1: Same as Fig.2a of the SM, but with Adam optimizer used for both the CNN and the eFCNs (initial learning rate of 0.001).

45 Motivated by the reviewer's comment, we repeated our numerical experiments using the adaptive Adam optimizer to
46 circumvent this question. Fig. 1 shows the results obtained in that case and confirms our conclusions. We will include
47 this additional finding in the revision.

48 In Ba & Caruana 2013, the shallow (and fully-connected) model is trained by regressing a previously trained deep (and
49 convolutional) model, whereas in our case the fully-connected models benefit from the architectural bias only through
50 the initial stages of training. We thank the reviewer for the pointer. We will add it to our discussion of papers related to
51 model compression.

[Meta-Review · NeurIPS 2019]

This paper examines the role of the architectural bias of CNNs, showing that the structural prior of locality and weight-sharing can improve performance of FCNs. The reviewers found the analysis novel and interesting and I agree with their consensus that this paper should be accepted.